# Modulation of Bone and Marrow Niche by Cholesterol

**DOI:** 10.3390/nu11061394

**Published:** 2019-06-21

**Authors:** Wenzhen Yin, Ziru Li, Weizhen Zhang

**Affiliations:** 1Department of Physiology and Pathophysiology, Peking University Health Science Center, Beijing 100191, China; 1811110036@pku.edu.cn; 2Department of Molecular & Integrative Physiology, University of Michigan Medical School, Ann Arbor, MI 48105, USA; 3Department of Surgery, University of Michigan Medical School, Ann Arbor, MI 48109-2800, USA

**Keywords:** cholesterol, bone metabolism, hematopoiesis, bone marrow adipocytes

## Abstract

Bone is a complex tissue composing of mineralized bone, bone cells, hematopoietic cells, marrow adipocytes, and supportive stromal cells. The homeostasis of bone and marrow niche is dynamically regulated by nutrients. The positive correlation between cardiovascular disease and osteoporosis risk suggests a close relationship between hyperlipidemia and/or hypercholesterolemia and the bone metabolism. Cholesterol and its metabolites influence the bone homeostasis through modulating the differentiation and activation of osteoblasts and osteoclasts. The effects of cholesterol on hematopoietic stem cells, including proliferation, migration, and differentiation, are also well-documented and further relate to atherosclerotic lesions. Correlation between circulating cholesterol and bone marrow adipocytes remains elusive, which seems opposite to its effects on osteoblasts. Epidemiological evidence has demonstrated that cholesterol deteriorates or benefits bone metabolism depending on the types, such as low-density lipoprotein (LDL) or high-density lipoprotein (HDL) cholesterol. In this review, we will summarize the latest progress of how cholesterol regulates bone metabolism and bone marrow microenvironment, including the hematopoiesis and marrow adiposity. Elucidation of these association and factors is of great importance in developing therapeutic options for bone related diseases under hypercholesterolemic conditions.

## 1. Introduction

Bone is a rigid organ but plays important roles in our body. It provides mechanical support for the soft tissues and enables mobility. It is also a mineral reservoir to endorse the calcium and phosphate homeostasis in body fluids. Bone and marrow niche are highly innervated and vascularized. Therefore, this seemingly closed system is tightly connected with the whole body metabolic homeostasis and subject to dynamical regulation by hormones and nutrients. Plenty of epidemiological studies have shown a positive correlation between cardiovascular disease and osteoporosis risk [1,2,3,4], suggesting a close relationship between hyperlipidemia and/or hypercholesterolemia and bone metabolism.

Cholesterol is an essential structural component of all animal cell membranes, and also contributes to synthesis of bile acids and steroid hormones [5,6]. Cholesterol, per se, is hydrophilic and mainly transported by lipoproteins in circulation. Depending on the density, lipoprotein particles are classified as chylomicrons, very-low-density lipoprotein (VLDL), intermediate-density lipoprotein (IDL), low-density lipoprotein (LDL)/"bad" cholesterol, and high-density lipoprotein (HDL)/"good" cholesterol. Cholesterol is essential for the body at normal levels, which could also become a silent danger for cardiovascular diseases when the circulating concentration is high. Hypercholesterolemia is usually a consequence of combination of environmental (over-weight, unhealthy diet, stress) and genetic susceptibility [7,8]. Even though a positive correlation between cholesterol intake and its serum concentrations has been noticed under some cases [9], the amount of cholesterol circulating in the blood is more greatly influenced by other nutrients, including saturated fatty acids, trans fatty acids, dietary soluble fiber and fructose to varying degrees [10,11].

Hypercholesterolemia links hematopoiesis with cardiovascular disease [12] and is associated with expansion of myeloid cells in the peripheral blood [13]. Statins are commonly used to lower cholesterol, by competitively inhibiting 3-hydroxy-3-methylglutaryl-CoA (HMG-CoA) reductase, the rate-limiting enzyme of the mevalonate pathway. They could lower LDL cholesterol and raise HDL cholesterol concentrations [14]. Interestingly, these cholesterol-lowering drugs (statins) have been reported to play protective roles in bone metabolism [15]. 

Although connections between hypercholesterolemia and hematopoiesis and bone homeostasis have been relatively well-documented, limited studies link the cholesterol with bone marrow adipocyte tissue (BMAT), which takes up ~50%–70% of the bone marrow volume in an adult [16]. BMAT is gaining more attention due to its close relationship with bone metabolism and hematopoiesis. In newborns, all these bones are filled with hematopoietic cells, but at later stages of life, the number of hematopoietic components decreases, and the bone marrow adipocytes increase [17,18,19]. The regulation of this dynamic change between marrow adipocytes and hematopoietic cells, and whether the change in marrow niche influences bone homeostasis are still unclear. Here we review the bone tissue as a complex system, which includes bone cells, hematopoietic cells, and marrow adipocytes, and summarize the latest insights about roles of cholesterol in bone and marrow niche homeostasis.

## 2. Cholesterol and Bone Homeostasis

Bone metabolism is a continuous cycle of bone formation and resorption, coordinated by osteoblasts, osteocytes, and osteoclasts [20,21,22,23,24]. Osteoblasts are mononucleate bone-forming cells, which are derived from mesenchymal stem cells and are responsible for making new bone and repairing older bone. Osteoblasts produce osteoid, a mixture of proteins, which is subsequently mineralized and becomes bone. Osteocytes are derived from osteoblasts that are incorporated into the newly formed osteoid and eventually become bone cells embedded in calcified bone. Osteoclasts are large multinucleated cells derived from a monocyte stem-cell lineage [25,26,27], which are in charge of bone resorption, and also play roles in global calcium homeostasis [28]. An imbalance in the regulation of bone resorption and formation results in many metabolic bone diseases including osteoporosis. 

In recent years, the association between serum lipid and bone metabolism has gained considerable interest [29,30,31,32,33]. Clinical studies have found that, in a large group of postmenopausal women, serum total cholesterol and LDL levels were negatively associated with bone mineral density (BMD) at all measured sites [34,35,36]. In rats, high cholesterol diet significantly decreased BMD and serum concentrations of osteoblastic markers (i.e., alkaline phosphatase, and osteocalcin), however the level of bone resorption marker (i.e., C-terminal telopeptide of type 1 collagen) was increased [37]. This indicates that high cholesterol is associated with increased bone resorption and reduced bone formation. Here we review the connections between cholesterol and osteoblast/osteoclast differentiation and further highlight the effects of cholesterol-reducing medication, statins, on bone homeostasis.

The complex relationship between cholesterol and bone formation (osteoblasts and osteocytes). There have been studies on cells, animal models, and clinical studies to assess the effects of serum cholesterol and/or cellular cholesterol on osteoblast cell proliferation and differentiation. Cholesterol inhibited the proliferation and differentiation of mouse osteoblast MC3T3-E cells in a dosage-dependent manner [37]. Cholesterol treatment reduced the expression of osteoblastic genes, such as alkaline phosphatase (ALP, *Alpl*), collagen 1 (*Col1a1*), bone morphogenic protein (*BMP2*) and runt related transcription factor 2 (*Runx2*). These findings suggest that free cholesterol may inhibit the BMP2 to block the expression of *Runx2*, *Alpl*, and *Col1a1* in osteoblast cells, which in turn inhibits osteoblast differentiation. Consistent with the in vitro studies, circulating concentrations of ALP and oscteocalcin were decreased in rats with 3% cholesterol diet for 3 months, which may contribute to the lower BMD. The gene profile analysis indicates that TGF-β/BMP2 and Wnt signaling pathway were inhibited by cholesterol diet [37]. TGF-β/BMP has an irreplaceable role in bone formation during mammalian development. Similarly, the Wnt signal almost regulates all aspects of osteoblast function [38,39]. Again, an atherogenic diet (high-fat/high-cholesterol with sodium cholate) fed to mice led to significantly reduced femoral mineral content and mineral density when compared with a low-fat/no-cholesterol diet fed animals [40]. Interestingly, in the recent study, Li et al. found that cholesterol plays a dual role in osteoblast differentiation through hedgehog-dependent and independent mechanisms in cultured ST2 cells, a bone marrow stromal cell line [41]. This study shows that exogenous cholesterol inhibits osteoblast differentiation, and physiological levels of endogenous cholesterol are essential for bone marrow stem cell osteogenesis. These studies suggest that the effect of cholesterol on osteogenesis is more complex than either “bad” or “good.”

Consistent with the essential function of endogenous cholesterol in promoting osteoblastic differentiation, inhibition of its biosynthetic pathway by targeting HMG-CoA (3-hydroxy-3-methylglutaryl coenzyme A) reductase, the rate-limiting enzyme in cholesterol biosynthesis, reduced ALP activity and expression, as well as mineralization, without changing *Runx2*, *Col1a1*, and osteocalcin (*Bgalp*) expression [42]. These results suggest that products of the cholesterol biosynthetic pathway are important for proper development of marrow stromal cells (MSCs) into functional osteoblastic cells and the capability of forming a mineralized matrix. Loading cholesterol complexed to methyl-β-cyclodextrin (MβCD) to MSCs greatly increased the cholesterol ester (CE) production, and promoted cell proliferation and osteoblastic differentiation, which involves the upregulation of *BMP2* and *Runx2*. These upregulated factors have positive effects on ALP activity and mineralized nodule formation, thus stimulating MSC osteoblastic differentiation. Furthermore, suppression of acyl-CoA:cholesterol acyltransferase (ACAT), which is the key enzyme for the esterification of free cholesterol to CE, by either inhibitor or SiRNA, reduced the extent for osteogenesis, indicating the osteogenic potency of cholesterol at this scenario was mostly due to CE levels [43].

Moreover, specific oxysterols, namely 22(R)-, 20(S)-, and 22(S)-hydroxycholesterol, products of cholesterol oxidation, have roles in pro-osteogenic and anti-adipogenic of pluripotent mesenchymal cells [44]. The combination of 22R + 20S or 22S + 20S oxysterols increased ALP activity, robust mineralization, and *Bgalp* gene expression in cultured M2-10B4 MSCs. In addition, these specific oxysterols act in synergy with BMP2 to induce osteogenic differentiation. These pro-osteogenic effects of specific oxysterols were partially mediated by extracellular signal-regulated kinase (ERK) and enzymes in the arachidonic acid metabolic pathway, i.e., cyclo-oxygenase (COX) and phospholipase A_2_ (PLA_2_) [44]. 

As mentioned above, circulating cholesterol is mainly transported by lipoprotein, and LDL-cholesterol is generally a negative factor for health. The circulating levels of LDL-cholesterol was significantly and negatively correlated with whole body BMD [45]. A 20-year-long prospective study involving 1396 men and women showed that the longer the duration of high serum total cholesterol level was, the more significant it became as a risk factor for any osteoporotic fractures [46]. In vitro studies on MSCs also demonstrated that minimal oxidized low-density lipoprotein (MM-LDL), and other bioactive oxidized lipids inhibit osteoblast differentiation by inhibiting ALP activity, collagen processing and mineralization, via mitogen-activated protein kinase-dependent pathway [47]. 

In contrast to LDL, HDL, which is called the "good" cholesterol, is a vital constituent of the lipoprotein transport system, regulating plasma and tissue lipid metabolism and homeostasis [48]. Addition of HDL can eliminate the effect of oxidized LDL on apoptosis of osteoblasts [49]. Apolipoprotein A-I (ApoA-I) is the major protein component of HDL particles in plasma. Consistent with the protective effects of HDL on bone metabolism, *ApoA-I* deficient mice have greatly reduced bone mass, which is mainly due to the defect in bone formation based on the static and dynamic histomorphometric analysis. *ApoA-I* deficiency also causes the impaired biochemical composition and biomechanical properties. Moreover, MSCs from the *ApoA-I* deficient mice showed reduced osteoblastic differentiation, and increased adipogenesis [50]. Overall, the effects of cholesterol on bone metabolism may be beneficial or unfavorable depending on the amount of supply, combination with other lipids, modification during metabolism and forms in circulation. Of note, the variety of research models on either animals or cell culture may also contribute the discrepancies between studies. 

Promotion of osteoclastogenesis by LDL-cholesterol. Osteoclasts are terminally differentiated, multinucleated cells formed by the fusion of mononuclear progenitors of the monocyte/macrophage family [51]. Osteoclasts release H^+^ ions to dissolve cement material and secrete various proteolytic enzymes to break down the bone matrix. Osteoporosis is generally caused by the imbalance between osteoblasts and osteoclasts. Pelton et al. found that diet-induced hypercholesterolemia is associated with an increase of circulating osteoclast activity marker, pyridinoline cross-links of collagen I fragment. Histomorphometric analysis revealed a significant increase in the number of osteoclasts present in the bones of the hypercholesterolemic versus the normocholesterolemic diet groups, which indicates that diet-induced hypercholesterolemia increases osteoclastogenesis, leading to decreases in BMD, bone volume fraction and number of trabeculae, and an increase in trabecular spacing [52]. In the rat model, hyperlipidemia induced by a high cholesterol diet reduced alveolar bone density and increased the number of tartrate-resistant acid phosphatase (TRAP)-positive osteoclasts. Interestingly, intake of vitamin C partially restored the alveolar bone density and osteoclast changes induced by a high cholesterol diet [53]. Vitamin C could reduce oxidative damage, promote osteoblastic differentiation and increase the production of type I collagen [54]. Therefore, reduction of oxidative stress by the use of antioxidants may suppress alveolar bone resorption and promote bone formation.

Intracellular cholesterol homeostasis is strictly controlled by exogenous cholesterol uptake and its intracellular de novo biosynthesis. Osteoclast lineage cells have very low expression levels of HMG-CoA reductase (*HMGCR*) [55], and its expression level is not up-regulated upon depletion of cholesterol from the plasma membrane [56]. Therefore, the uptake of exogenous cholesterol plays a more important role in regulating osteoclast differentiation than its de novo biosynthesis. Cholesterol delivery via LDL significantly increased osteoclast viability. Removal of cholesterol in osteoclast via HDL or cyclodextrin treatment dose-dependently induced apoptosis. Consistent with the role of LDL receptor (LDLR) in cholesterol endocytosis, *Ldlr-/-* mice increased bone mass [56]. This change was accompanied by a decrease in bone resorption parameters and the number of osteoclasts, while the bone formation parameters remained unaltered. Furthermore, the size and lifespan of osteoclast isolated from *Ldlr-/-* mice were decreased, and it was prone to spontaneously induce apoptosis. In addition, *Ldlr*-deficient osteoclast precursor differentiation is significantly delayed and contains fewer nuclei. All these findings indicate that osteoclastogenesis in vitro is highly dependent on exogenous LDL, and *Ldlr* deficiency impairs osteoclast formation and decreases cell fusion [56,57]. 

Studies from co-culture of mouse spleen cells, which could develop into multinucleated osteoclast-like cells, with TMS-14 cells mouse stromal cell line, which is unique in supporting the differentiation of osteoclast-like cells, in the LDL-deficient serum, suggest that cholesterol plays an important role in cell-cell fusion events that occur during osteoclast formation [58]. One possible mechanism for this defect in cell fusion is the abnormality of the signal transduction pathway necessary for osteoclast formation, such as overactivation of ERK1/2 and insensitivity of Akt to RANKL (Receptor activator of nuclear factor kappa-Β ligand) stimulation, which is important for osteoclast differentiation and function [56,58]. Moreover, osteoclastogenesis is also highly dependent on the integrity of lipid rafts and extracellular lipoproteins [59], which suggests possible involvement of cholesterol.

Opposite to LDL, HDL induces the apoptosis of osteoclasts. These effects may attribute to the up-regulation in the expression of *ABCG1* (ATP-binding cassette sub-family G member 1) and cholesterol efflux from osteoclasts, which impair cholesterol homeostasis in osteoclasts [60]. In addition, liver x receptor (LXRα, β) is also responsible for regulating cholesterol homeostasis in cells. Both LXRs affect cell function in the bone. LXRα has an effect on osteoclast activity, mainly in cortical bone, while LXRβ regulates trabecular bone turnover [61].

Beneficial effects of statins on bone metabolism. Statins are cholesterol-reducing medication which competitively inhibits the activity of HMG-CoA reductase, a rate limiting enzyme of the cholesterol biosynthesis pathway in a reversible fashion [62]. Clinical and animal studies have evidenced the active role of statins in reducing the risk of fracture [15,63,64,65], by promoting bone formation and concurrently inhibiting osteolytic metastasis [66,67,68,69]. The osteoprotective effects of statins seem to be more prominent with a dependency on the cumulative dosage and statin intensity [70].

The effect of statins on bone metabolism was first discovered by screening for agents that activate the mouse BMP2, a protein essential for bone formation [71]. Statins are classified as lipophilic (atorvastatin, simvastatin and lovastatin) and relatively hydrophilic (pravastatin and rosuvastatin) according to their inherent polar characteristics. Individual bone effects may vary due to differences in their inherent polarity and bone bioavailability. Substantial clinical trials data demonstrate that only lipophilic statins significantly enhanced *BMP2* expression to promote osteoblast differentiation [72]. Mundy et al. injected statins (lovastatin and simvastatin) subcutaneously over the calvaria of mice and found marked increases in the amount of bone formation in treated animals [71]. Studies in vitro clearly demonstrate that statins promote osteoblast differentiation as evidenced by stimulating expression of *BMP2*, vascular endothelial growth factor (*VEGF*), osteocalcin, and bone sialoprotein (*BSP*), and enhanced mineralization [73,74]. This stimulating effect of statins on bone formation evidenced by in vitro studies suggest that the benefits of statins may not only be due to lipid-lowering effects but also due to their pleiotropic effects. 

An elegant series of experiments have partially explained the mechanisms of pleiotropic osteoprotective effects of statins. For instance, statins induced osteogenesis and bone formation. The statin-mediated activation of the *BMP2* promoter can be abolished by the addition of mevalonate, the downstream metabolite of HMG-CoA reductase, suggesting that it is a result of HMG-CoA reductase inhibition [75]. Statins inhibited the production of osteoclasts by down-regulating the expression of *RANKL* and up-regulating the expression of osteoprotegerin (OPG, *Tnfrsf11b*), which inhibits osteoclastogenesis by binding with RANKL and thus preventing its interaction with RANK on osteoclast precursors [76]. Although the side effects of statins on muscle and liver are noticed, the majority of studies show positive results on the usage of statins in clinical patients. Thus, statins could be considered as one of the potential preventive or therapeutic options for osteoporosis in the future. 

## 3. Cholesterol and Hematopoietic Stem Cell Proliferation, Mobilization, and Hematopoiesis

One of the major functions of bone marrow is to supply mature blood cells into circulation, such as red blood cells to carry oxygen, lymphocytes to anchor the adaptive immune system, and the myeloid lineage cells, which are involved in innate immunity and blood clotting. In addition to the tight association of hypercholesterolemia with cardiovascular disease [12], the cholesterol content is closely related to the homeostasis of the bone marrow microenvironment, and further influence the hematopoiesis.

Hematopoietic stem cells (HSCs) are characterized by their ability for self-renewal and generation of all mature blood cells. The balance between self-renewal and differentiation is crucial to maintain the integrity of the entire hematopoietic system. Quiescence is thought to be a fundamental characteristic of HSCs, the precise regulation of their cell cycle is required for avoiding stem cell exhaustion, but still effectively producing mature hematopoietic cells [77]. The decision of whether or not to exit quiescence is considered to be influenced by both cell-intrinsic and -extrinsic signals induced by various stresses. 

Gu et al. found that LDL-cholesterol levels are correlated with HSC frequency in circulation in healthy volunteers [78]. Hypercholesterolemic patients mobilize more HSCs to circulation after treatment with cyclophosphamide and granulocyte colony-stimulating factor (G-CSF) [79]. Mice with high cholesterol levels have also shown that HSCs are induced to mobilize and proliferate, thereby increasing the number of neutrophils, lymphocytes, and hematopoietic stem and progenitor cells (HSPCs) in the circulation [80]. Consistently, another study by Lang et al. showed that statin treatments reduced the quantity of CD34^+^ HSPCs, and levels of which positively correlated with serum LDL-cholesterol levels [81]. These findings indicate that cholesterol has a stimulating effect on HSPCs. Mechanistically, LDL-cholesterol increases the number of CD34^+^ HSPCs in the bloodstream by increasing proliferation, and by stimulating the levels of mobilizing cytokines interleukin-17 (IL-17) and granulocyte colony-stimulating factor (G-CSF). 

Cholesterol efflux was associated with IL17/G-CSF signaling axis controlling HSPC mobilization. Deletion of genes related to cholesterol efflux pathways is helpful to understand how cholesterol impacts the biology of hematopoietic stem cells [81]. Abca1, ATP-binding cassette transporter, transports cholesterol from membranes to nascent HDL. Abcg1, the ATP-binding cassette sub-family G member 1, transports cholesterol to mature HDL [12,82,83]. The *Abca1-/-Abcg1-/-* mice showed an expansion and proliferation of HSPCs in BM and significant leukocytosis [84]. Further studies indicate that the production of IL-23, which drives the IL-17 producing cells, is increased in all BM cells or macrophages and dendritic cells lacking *Abca1* and *Abcg1* [84]. Apoe is expressed on the surface of HSCs and interacts with Abca1 and Abcg1 to promote cholesterol efflux [85]. Expansion and proliferation of HSCs have also been shown to exist in *Apoe-/-* mice [86]. Inversely, increased levels of HDL can prevent or reverse HSC mobilization and extramedullary hematopoiesis in *Abca1-/- Abcg1-/-* mice and *Apoe-/-*mice [84]. Consistently, overexpression of the human *ApoA-I* transgene to increase HDL levels also prevent myeloproliferative diseases in *Abca1-/- Abcg1-/-* mice [87]. All these studies suggest that cholesterol efflux pathways are important in the control of HSC mobilization, which may translate into therapeutic strategies for atherosclerosis and hematologic malignancies.

In addition to the effects of cholesterol on HSC homeostasis in bone marrow, epidemiological studies have also found that high cholesterol levels usually have higher circulating monocytes and neutrophils in humans [88]. High fat/high cholesterol diet causes thrombocytosis and lymphocytosis in mice [80]. *Abca1-/-Abcg1-/-* mice showed a significant myeloproliferative phenotype of monocytes, neutrophils and eosinophilia [87]. In addition to hypercholesterolemia, *Apoe-/-* mice and *Ldlr-/-* mice also exhibit mononucleosis and neutropenia [86,89,90]. Wilhelm et al. reported that *ApoA-I* deficiency leads to cholesterol enrichment in lymph nodes and, more importantly, a significant increase in T cell proliferation and activation [91]. Cholesterol metabolites, such as androgens and estrogens, negatively regulate B lymphocyte production [92,93,94,95]. In contrast, epidemiological investigations found that children with the lowest HDL levels had higher peripheral blood mononuclear cell counts, and HDL levels were negatively correlated with percentage of monocytes [96].

Platelets play an important role in all stages of the development of atherosclerotic lesions [97]. ABCG4, expressed in the megakaryocyte progenitor (MkP) population of bone marrow, promotes cholesterol efflux to HDL. With the decrease in cholesterol efflux caused by *Abcg4* deficiency, *Abcg4-/-* mice exhibit MkP proliferation and expansion, thrombocytosis, increased platelet/leukocyte aggregation and accelerated atherosclerosis [98]. Leukocytosis is also a risk factor for human atherothrombotic disease [98]. Moreover, Abca1 exerts anti-inflammatory activity by regulating the cholesterol content of membrane lipid rafts. By modulating lipid raft organization on the cell surface and inhibiting inflammation, macrophage Abca1 can prevent atherosclerosis by removing excess lipids from macrophages [99]. ABC transporter deficiency causes cholesterol restriction in cells, further destroying the steady-state of cholesterol, eventually leading to excessive inflammation of macrophages and atherosclerosis. Therefore, cholesterol efflux plays important roles in HSC stabilization and development of atherosclerotic lesions. Development of therapeutic strategies by increasing cholesterol efflux, such as infusions of cholesterol poor reconstituted HDL and upregulation of Abca1/Abcg1 treatment, may provide potentials to decrease coronary heart disease risk [100] with concurrent benefit for bone homeostasis.

## 4. Cholesterol and Bone Marrow Adiposity

Bone marrow adipocytes (BMAs) originate from MSCs. These cells may contribute to the regulation of bone homeostasis [101,102] through the direct or indirect interactions with osteoblasts and osteoclasts during bone remodeling [103]. A considerable body of experimental evidence derived from cell culture and animal studies indicates that BMAs increases in states of osteoporosis [104], caloric restriction [105], aging [106] and anti-diabetes therapies [107]. Since MSCs are the common progenitors for both osteoblasts and adipocytes [108,109,110], they perform like a playground ‘‘see-saw’’ that can swing back and forth between osteogenesis and adipogenesis [111]. Therefore, it has been suggested that in a variety of types of osteoporosis, adipogenic differentiation increases at the expense of osteogenic differentiation [47,112]. This may result in an increase in the number of adipocytes and the decrease of osteoblasts. 

Bone marrow adipose tissue (BMAT) is distinct from white- and brown- adipose tissues in morphology, gene expression profile and regulation mechanisms. Only a few studies connected global lipid metabolism with BMAT. Slade et al. reported that in type 1-diabetes, there was a positive correlation between bone marrow adiposity and serum lipid levels [113]. In a healthy population, HDL-cholesterol levels were inversely associated with bone marrow fat content, independent of BMI, age, and exercise status [114]. As briefly mentioned above, oxysterols are involved in MSC differentiation with effects on inhibiting the formation and differentiation towards fat cells [44,115]. 

20(*S*)-hydroxycholesterol is the most potent naturally oxysterol [116], which has been shown the potency to induce osteogenic- and inhibit adipogenic-differentiation of murine MSCs [44,116,117]. Consistent with the pro-osteogenic function of 20S, it inhibits the MSCs differentiation towards adipocytes via the hedgehog signaling pathway [116,117,118]. The inhibition of adipogenesis by 20S is also associated with a decrease of peroxisome proliferator-activated receptor gamma (*PPARγ*) mRNA expression, which is a key regulator of adipogenic differentiation. Therefore, *PPARγ* may be a potential target for oxysterol-induced MSC transformation from the adipogenic- to the osteogenic-pathway [116]. Moreover, oxysterol induces sustained activation of ERK pathway, which is a negative regulator of MSC differentiation into adipocytes [119]. However, ERK activation appears to be a necessary but not sufficient step to regulate osteogenesis [44]. 

As described above, statins can not only promote osteogenesis, but also inhibit the differentiation of fat cells. It therefore has the potential to treat common metabolic bone diseases such as senile osteoporosis. Simvastatin enhanced ALP activity, osteogenic gene expression, and mineralization. Simultaneously, simvastatin also decreased Oil Red O staining and inhibited the gene expression of lipoprotein lipase (*LPL*) and *PPARγ* in a dose-dependent fashion. These results indicate that simvastatin has anabolic effects on bone which might associate with the inhibition of adipocytic differentiation [120]. Opposite to the effects of LDL-lowing statins, perturbation of HDL by using *ApoA-I-/-* mice that lack classical ApoA-I containing HDL, or *LCAT-/-* mice that have only immature HDL and relatively reduced HDL-cholesterol levels, significantly elevated number of adipocytes in the cancellous bone marrow when mice were fed with western diet [121]. Although the effects of oxysterol and HDL on BMAs are partially revealed, more studies are needed in the future to reveal the effects of cholesterol and its metabolites on marrow adiposity, which may play important roles in bone and hematopoietic homeostasis. 

## 5. Summary and Future Directions

Cholesterol and its metabolites have complex functions in osteogenesis, osteoclastogenesis, and bone homeostasis depending on the forms of cholesterol. In addition to the cholesterol-lowering effects, some specific statins also possess pleiotropic effects and benefit bone metabolism, which is supported by most of the studies performed in animals or cell culture. However, the relationship between cholesterol and bone metabolism is confounded in human studies. Therefore, the cholesterol-bone relationship should be fully studied and understood in more populations. More importantly, we have very little knowledge about the mechanisms that explain the association and communication factors between cholesterol and bone. Understanding these associations will greatly further our understanding about the clinical relationship between hypercholesterolemia and bone metabolism, and may develop novel treatment options for bone diseases.

The hematopoietic cells and bone marrow adipocytes take up the majority of space in bone cavity. It is relatively well-documented that cholesterol influences the HSC proliferation and mobilization. In addition, cholesterol drives hematopoiesis towards meyloid lineage, which potentially contributes to the development of atherothrombosis. However, the relationship between cholesterol and marrow adiposity remains elusive. The functions of BMAT are still largely unknown, which may play important roles in local bone marrow niche or produce unique secretions, and further influence the function of bone cells and HSPCs. Future research by exploring the physiological and pathological functions of BMAT will provide us deep insights about the importance of bone marrow niche homeostasis in local hematopoiesis and osteogenesis, which may potentially benefit the therapeutic strategies for atherothrombosis and osteoporosis (Figure 1).

## Figures and Tables

**Figure 1 nutrients-11-01394-f001:**
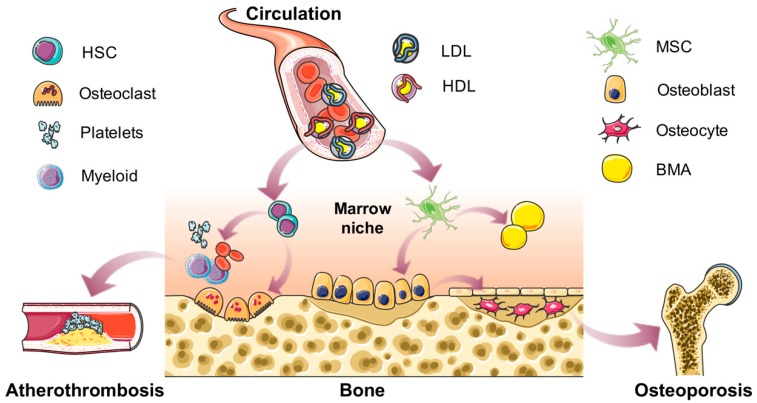
Summary of the effects of cholesterol on bone and marrow niche homeostasis. In the circulation, cholesterol is present in the forms of high-density lipoprotein (HDL), low-density lipoprotein (LDL), and very-low-density lipoprotein (VLDL) with lipoproteins. Normal concentrations of cholesterol are essential for the body. However, when the circulating concentration changes, it may become a risk factor. This review focuses on the roles of cholesterol in bone metabolism and bone marrow nice homeostasis. Cholesterol could drive the differentiation of marrow stromal cells (MSCs) into either osteoblasts or bone marrow adipocytes (BMAs), and further influence the bone mass or osteoporosis. Meanwhile, cholesterol affects the proliferation and mobilization of hematopoietic stem cells (HSCs), myeloid lineage differentiation and thrombocytosis, effects of which may potentially contribute to the development of atherothrombosis.

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
