# Peer review of "Modulation of Bone and Marrow Niche by Cholesterol"

_nutrients, 2019, doi:10.3390/nu11061394_

Round 1
Reviewer 1 Report
The review manuscript of Yin et al. describes the role of cholesterol in the function and homeostasis of bone. The manuscript was a pleasure to read with many important details. The literature review was complete. I have only two small comments.
There are only a few minor grammatical errors that will need to be corrected.
The authors have done a good job of summarizing the literature. The authors could take advantage of this summary to make a few conclusions, in effect, to bring us to a new insight. For example, I think it is safe for the authors to conclude that statins are beneficial for bone growth and maintenance. In the era of side effects and lawsuits, this may be the most important conclusion.
Author Response
The review manuscript of Yin et al. describes the role of cholesterol in the function and homeostasis of bone. The manuscript was a pleasure to read with many important details. The literature review was complete. I have only two small comments.
Reply: Thank you!
There are only a few minor grammatical errors that will need to be corrected.
Reply: We have gone through the whole manuscript for couple more times and corrected the grammatical errors as we noticed.
The authors have done a good job of summarizing the literature. The authors could take advantage of this summary to make a few conclusions, in effect, to bring us to a new insight. For example, I think it is safe for the authors to conclude that statins are beneficial for bone growth and maintenance. In the era of side effects and lawsuits, this may be the most important conclusion.
Reply: Thank you for the great suggestion. We added relative conclusions and insights according to the reviewed literatures in lines 193-219; 233-236; 342-344; 478-482.
Reviewer 2 Report
The review entitled Modulation of bone and marrow niche by cholesterol by Wenzhen Yin and collaborators summarizes the latest progress of how cholesterol regulates bone metabolism and bone marrow microenvironment, including the hematopoiesis and marrow adiposity.
The review is interesting and sounds good, but it would be important to add how nutrition could impact this processes. This will help to let the paper falls in the scope of the journal.
Author Response
The review entitled Modulation of bone and marrow niche by cholesterol by Wenzhen Yin and collaborators summarizes the latest progress of how cholesterol regulates bone metabolism and bone marrow microenvironment, including the hematopoiesis and marrow adiposity.
The review is interesting and sounds good, but it would be important to add how nutrition could impact this processes. This will help to let the paper falls in the scope of the journal.
Reply: Thank you for the suggestion. We have now addressed the connection between the cholesterol with nutrients in lines 44-65.